# Early and Objective Evaluation of the Therapeutic Effects of ADHD Medication through Movement Analysis Using Video Recording Pixel Subtraction

**DOI:** 10.3390/ijerph19063163

**Published:** 2022-03-08

**Authors:** Ying-Han Lee, Chen-Sen Ouyang, Yi-Hung Chiu, Ching-Tai Chiang, Rong-Ching Wu, Rei-Cheng Yang, Lung-Chang Lin

**Affiliations:** 1Department of Post Baccalaureate Medicine, Kaohsiung Medical University, Kaohsiung 807, Taiwan; yinghang99@gmail.com; 2Department of Information Engineering, I-Shou University, Kaohsiung 840, Taiwan; ouyangcs@isu.edu.tw (C.-S.O.); asd456987321@gmail.com (Y.-H.C.); 3Department of Computer and Communication, National Pingtung University, Pingtung 912, Taiwan; cctai@mail.nptu.edu.tw; 4Department of Electrical Engineering, I-Shou University, Kaohsiung 840, Taiwan; rcwu@isu.edu.tw; 5Department of Pediatrics, Kaohsiung Medical University Hospital, Kaohsiung Medical University, Kaohsiung 807, Taiwan; 6Department of Pediatrics, School of Medicine, College of Medicine, Kaohsiung Medical University, Kaohsiung 807, Taiwan

**Keywords:** attention-deficit/hyperactivity disorder, video analysis, pixel subtraction, Swanson, Nolan, and Pelham questionnaire

## Abstract

Attention-deficit/hyperactivity disorder (ADHD) affects approximately 5–7% of school-age children. ADHD is usually marked by an ongoing pattern of inattention or hyperactivity–impulsivity, leading to functioning or developmental problems. A common ADHD assessment tool is the Swanson, Nolan, and Pelham (SNAP) questionnaire. However, such scales provide only a subjective perspective, and most of them are used to evaluate therapeutic effects at least 3–12 months after medication initiation. Therefore, we employed an objective assessment method to provide more accurate evaluations of therapeutic effects in 25 children with ADHD (23 boys and 2 girls). To evaluate the participants’ improvement and treatment’s effectiveness, the pixel subtraction technique was used in video analysis. We compared the efficacy of 1-month Ritalin or Concerta treatment by evaluating the movement in each video within 3 h of medication administration. The movement value was defined as the result of a calculation when using the pixel subtraction technique. Based on behavior observation and SNAP scores, both parent- and teacher-reported scores decreased after 1 month of medication (reduction rates: 19.61% and 16.38%, respectively). Specifically, the parent-reported hyperactivity subscale and teacher-reported oppositional subscale decreased more significantly. By contrast, the reduction rate was 39.27%, as evaluated using the average movement value (AMV). Considering symptomatic improvement as a >25% reduction in scores, the result revealed that the AMV decreased in 18 patients (72%) compared with only 44% and 56% of patients based on parent- and teacher-reported hyperactivity subscale scores. In conclusion, the pixel subtraction method can serve as an objective and reliable evaluation of the therapeutic effects of ADHD medication in the early stage.

## 1. Introduction

Attention-deficit/hyperactivity disorder (ADHD) is one of the most common childhood neurodevelopmental disorders and is characterized by an ongoing pattern of inattention or hyperactivity–impulsivity, leading to functioning or developmental problems [1]. ADHD is highly prevalent worldwide in children, adolescents, and even young adults, affecting 5–7% of children and adolescents, with a higher incidence in men (male:female ratio, approximately 3:1–4:1) [2].

ADHD can profoundly affect children’s academic achievements, well-being, and social interactions [2]. With the high incidence of many adverse functional outcomes in ADHD, diagnosis and treatment have become critical clinical issues [3]. Its treatment can involve pharmacological and nonpharmacological therapies and may achieve optimal coverage among individuals with ADHD. However, not every patient with ADHD responds well to treatment or achieves optimal symptom control. No consensus exists regarding the definition of treatment response to ADHD medications [2].

Symptomatic improvement in ADHD is a critical component for assessing the course of ADHD. The American Academy of Pediatrics 2019 guidelines mentioned that regardless of treatment, children with ADHD aged 4–18 demonstrated a 25% symptom reduction at 6–12 months from baseline, as measured using the Vanderbilt ADHD Diagnostic Rating Scale, which is considered a symptomatic improvement [4].

Diagnosing and monitoring ADHD, especially in children, remains challenging despite the use of various assessment tools, including the Swanson, Nolan, and Pelham (SNAP) questionnaire; the Vanderbilt ADHD Diagnostic Rating Scale; and the visual analog scale [5]. Among them, the SNAP questionnaire is the most popular and is composed of three subscales: inattention, hyperactivity–impulsivity, and oppositional defiant disorder symptoms. The assessment tools used are always based on ratings made by teachers or parents and diagnoses made only by specialists [1]. Thus, these scales provide only subjective perspectives and probably lead to biased diagnoses or evaluations. An objective tool is essential for assessing the treatment response among patients with ADHD and providing information regarding the optimal therapeutic effects of ADHD medications.

Image processing has attracted interest in video surveillance applications [6], especially for understanding human activity [7]. Movement detection involves tracking moving objects by using an algorithm [6]. Human movement analysis can serve as an objective assessment. A common movement detection method is pixel subtraction between two consecutive gray images after the separation of frames from a video. Pixel subtraction employs an algorithm that can detect movement within specific ranges for movement-tracking applications. It refers to the image difference or pixel–pixel difference between two consecutively obtained images. A difference within the threshold indicates movement of the subject. Because the pixels of the color images were three-dimensional, we converted them into grayscale images (Figure 1). This conversion decreased the operational calculation time without affecting the result of the movement analysis. Thus, it is used in various circumstances for movement detection [7]. However, this technique has been underutilized in the medical field. In a study on the application of image subtraction for magnetic resonance imaging (MRI), researchers claimed that regions of changes could be detected through the subtraction of pixels between two images, and in the case of multiple sclerosis lesions, they reported that lesions could be detected through the subtraction of MRI images with a gadolinium-diethylenetriamine pentaacetic acid (GdDTPA) contrast agent [8]. This offers insights into the implementation of such statistical techniques, particularly in medical applications.

Devi et al. mentioned that the frame subtraction method is an efficient alternative to comparing image pixel values in subsequent frames captured 2 s apart, with the first frame serving as reference and the second frame containing the moving object; the two frames (images) are compared to detect movement by calculating the differences in pixel values [7,9]. Increased activity is characteristic of patients with ADHD; thus, movement analysis can be a useful technique [10,11]. In the present study, we compared two frames in an input video by using movement detection in the first attempt to analyze the treatment response among patients with ADHD. We thus introduce the pixel subtraction method, a new objective approach for assessing the ADHD treatment response, which can serve as an adjuvant to the SNAP questionnaire.

## 2. Patients and Methods

### 2.1. Participants

In this study, the mean age of enrolled patients was 7 years 9 months ± 1 year 2 months, newly diagnosed as having ADHD by pediatric neurologists or psychiatrists; they were involved in the study for the treatment response. ADHD was diagnosed on the basis of the Diagnostic and Statistical Manual of Mental Disorders, Fifth Edition (DSM-V) criteria, and ADHD severity was assessed using the SNAP-IV tool. Exclusion criteria were a history of epilepsy, intellectual disability, drug abuse, head injury, or psychotic disorder. A family member or legal guardian of each patient provided informed written consent for their child’s participation. Ethical approval was obtained from the Institutional Review Board of Kaohsiung Medical University Hospital [KMUIRB-SV (I)-20190060].

### 2.2. Evaluation of the Therapeutic Effects of ADHD

After receiving the diagnosis of ADHD, the participants started their 1-month treatment with Ritalin (methylphenidate HCL, Norvatis, Basel, Switzerland) 10 mg/day (17 patients) or Concerta (methylphenidate HCL, Johnson & Johnson, New Brunswick, NJ, USA) 18 mg/day (8 patients). In addition, to monitor the drug compliance of the participants, self-reporting and medication measurement by parents were used to ensure medication regimen adherence. To ensure more precise monitoring of their response to the psychostimulants, they were asked to take morning doses after breakfast. Before the day of starting the psychostimulants, the behaviors of the participants were assessed using the following two methods: the SNAP-IV rating scales and movement detection using the pixel subtraction method. The SNAP-IV questionnaire was completed by both parents and teachers, either at home or school, and the pixel subtraction method required the capture of video footage during their first consultation by a pediatric neurologist. After 1 month of receiving the treatment with psychostimulants, they were assessed again with the same methods. The SNAP-IV questionnaire was completed by parents and teachers. During the day of consultation, they had taken the medicines before the visit within 3 h of medication administration.

The SNAP-IV comprises 26 items that are rated on a 4-point scale from 0 (not at all) to 3 (very much). The items are divided between three subscales: inattention (nine items), hyperactivity–impulsivity (nine items), and oppositional defiant disorder symptoms (eight items) specified in the DSM-V. Subscale scores are determined by calculating the average of the item scores per subscale. Each subscale score indicates the symptoms exhibited by the patients and helps in the categorization of the three ADHD subtypes: inattentive (ADHD-I; inattention symptoms and few or no hyperactivity symptoms), hyperactive–impulsive (ADHD-H; hyperactivity or impulsivity symptoms and few or no inattention symptoms), or combined (ADHD-C; both inattention and hyperactivity symptoms).

Movement detection by using the pixel subtraction method was performed by analyzing the captured videos that were obtained in the consultation room. We compared the efficacy of 1-month Ritalin or Concerta treatment by evaluating the movement in each video within 3 h of medication administration. The video recorder was placed in a fixed position in the consultation room and was unnoticed by the patients (Figure 2). In the present study, 4–6 min video recordings were obtained from each patient before and after treatment. For minimizing the biased comparison, only an initial four minutes video recording of each visit for each patient was used for analysis. The clinical settings and time of visit in the consultation room stayed as similar as possible. For example, the same doctor and nurse appeared in the room during video recording. All the patients visited for consultations and video recordings in the morning. They received clinical observations only in the room without any conversation or physical examination during video recordings.

According to the study by Wolraich et al., the symptomatic improvement in patients with ADHD was defined as a 25% reduction in symptoms from baseline, as measured using the Vanderbilt ADHD Diagnostic Rating Scale [4]. In the present study, we used the same criteria to evaluate therapeutic effects for symptoms or movements reduction.

Figure 3 presents a flowchart of the movement analysis. A recorded video was provided as input to the analysis. This input video was then converted into multiple frames of color images by using MATLAB, with two images obtained per second. For example, a 5-min video comprised 600 continuous images.

After obtaining a series of grayscale images in sequence, the first (Figure 1c) and second grayscale images (Figure 1d) served as the input to the pixel subtraction method, and the subtracted image (output) was then produced (Figure 1e). This step was repeated for the subsequent images; the second and third images were the input used to obtain the subtracted image. A series of subtracted images were thus obtained as outputs and were then further used in the movement analysis. The pixel subtraction of two images was performed in a single pass. The output pixel values were given by:(1)Q(i,j)=|P1(i,j)−P2(i,j)|
where *i* and *j* are the pixel x and y coordinates.

In our proposed method, when there was no movement difference, the pixel values of the two consecutive images were considered equal. Thus, the output pixel value was zero after pixel subtraction, and the pixel of the output image was displayed as black. By contrast, if there was any change or movement occurring between the two input images, the output pixel indicated movement through a light color. The light part of the subtracted image (Figure 1e) indicates the movement difference between the two input images. By using this pixel subtraction technique, we could easily identify small movements of our patients, which would usually be imperceptible to the naked eye.

Using this method, movement was identified and tracked only at the specific regions of movement or at selected areas of the subtracted images. We defined the selected area as the rectangular region of the participant’s movement (Figure 1e). To trace the movement based on the pixel values, we set a threshold (*θ*) for the pixel value. When the difference between pixel values, *Q*(*i*,*j*), exceeded the threshold *θ*, *D*(*i*,*j*) was set as 1; otherwise, it was set as 0. In dynamic image processing, all pixels in *D*(*i*,*j*) with the value of 1 were considered to be the result of object movement [12].
(2)$$\{\begin{array}{c} D\left(i,j\right)=1,\;if\;Q\left(i,j\right)>\theta \\ D\left(i,j\right)=0,else\; \end{array}$$

In the pixel subtraction technique, we excluded effects such as image noise or tiny movements of the patient while running the MATLAB analysis; the threshold pixel value in this paper was set as 100 (*θ* = 100), which was considered to indicate significant movement [13]. The sum of all the elements of *D*(*i*,*j*) provided the feature for the participants. This feature characterizes the movement of the patient in the entire video. A movement measure, *M*, was defined as the following equation:(3)$$M=\frac{\sum_{k=1}^{N}\sum_{\left(i,j\right)\in SA}^{}D^{k}\left(i,j\right)}{N\times L}\times 100\%$$
where $D^{k}\left(i,j\right)$ was the value of D(i,j) corresponding to the kth frame, SA was the set of image coordinates corresponding to the selected area, N was the number of frames of the video, and L was the number of pixels included in the selected area.

## 3. Statistical Analysis

Statistical analyses were conducted using SPSS (V20.0) (IBM, Armonk, NY, USA). Data are presented as the mean ± standard deviation. To analyze for any significant difference between the movement of the patient pre- and post-medication, we conducted a paired *t* test. *p* < 0.05 was set as statistically significant.

## 4. Results

We recruited 25 patients (23 boys and 2 girls) for analyzing the therapeutic effectiveness of ADHD medication by using SNAP-IV scores and movement analysis.

The average parent-reported SNAP scores before and after treatment were 43.6 ± 11.35 and 35.08 ± 13.14, respectively, indicating a significant reduction of 19.61% (*p* = 0.013), but the difference in teacher-reported scores (16.38%: 38.08 ± 17.09 before and 31.84 ± 18.08 after treatment) was not significant compared with the parents’ scores (*p* = 0.074).

The parent and teacher ratings for each subscale of the SNAP were compared with respect to treatment analysis in Table 1. Among all the subscale scores, only hyperactivity subscales from parents and oppositional subscales from teachers showed significant differences after the 1-month treatment with Ritalin or Concerta.

For the movement analysis, the average movement values (AMVs) of all patients before and after 1 month of treatment (0.9940 ± 0.7556 and 0.6036 ± 0.4405, respectively) indicated a significant difference (*p* = 0.001; Table 2).

On the basis of the result presented in Table 3 the results of the movement analysis revealed that the proportion of children who achieved symptomatic improvement, showing a reduction of more than 25% (*n* = 18, 18/25 = 72%), was higher than that obtained in the traditional approach using SNAP-IV scores. Using SNAP-IV rating scales by parents and teachers, fewer children showed an improvement trend. More than 50% of parents noticed that their children did not show a significant improvement after treatment. For instance, based on parents’ ratings in hyperactive and oppositional subscales, only 11 and 9 children, respectively, exhibited >25% reduction after treatment. Similarly, based on the teachers’ ratings in the hyperactive subscale, 14 exhibited symptomatic improvement (>25% reduction). The correlation coefficient between AMV and SNAP scores parent-reported was −0.144. The correlation coefficient between AMV and SNAP scores teacher-reported was −0.094. There was no significant correlation between AMV and SNAP scores. Regarding to the consistency between the treatment response evaluated using AMV and SNAP, the consistency between AMV and the hyperactivity subscale from parents was 88%; the consistency between AMV and the hyperactivity subscale from teachers was 68% by a kappa statistic measurement.

## 5. Discussion

In this study, the SNAP-IV scores for the hyperactivity subscale from parents and the oppositional subscale from teachers decreased significantly. By contrast, the reduction rate was 39.27%, as evaluated using the AMV. With improvement considered as a reduction of more than 25%, the results revealed that the AMV decreased in 18 patients (72%), of whom only 11 patients (44%) and 14 patients (56%) improved by more than 25% according to the SNAP hyperactivity subscales obtained from parents and teachers.

According the findings of the study, the average parent-reported SNAP scores indicated a significant reduction compared with teacher-reported scores. This indicates there were inconsistent reports between parents and teachers. The 26-item SNAP-IV has been widely used as a standard behavioral rating scale for the diagnostic assessment and core symptom management of ADHD in the current clinical setting, as defined by the DSM-V [14]. Although the SNAP scale is a widely used clinical and research tool, several issues exist regarding its validity and the measurement invariance between parent and teacher ratings [15,16]. When comparing the SNAP-IV subscales between parents and teachers, only parents’ and teachers’ ratings in respective hyperactive and oppositional variables demonstrated a significant score reduction after 1 month of ADHD treatment. Therefore, its specificity for ADHD subtypes might be an issue. Hall et al. reported that SNAP-IV scores were highly sensitive to detecting ADHD symptoms but that the specificity was particularly poor compared with clinical diagnoses [16]. Hall et al. explained that parents and teachers rate the same child’s behavior in the same way but to a different degree, which may be due to the different environments in which the two informants observe the child [16]. Our findings indicated that children are less hyperactive at home, as reported by parents, which is contrary to several reviews. The behavioral characteristics of ADHD differ between school and home; because school is a more structured environment, noncompliance regarding attentiveness may occur [17]. However, both Concerta and Ritalin reach their initial peak concentration within 1–2 h, followed by a gradual ascending increase in plasma levels for the achievement of a maximum plasma concentration at approximately 6–10 h for Concerta and 6 h for Ritalin [18]. A total of 17 children in our cohort were prescribed Ritalin. Therefore, the parents may notice symptomatic improvement in their children after taking the medicine within 2 h before leaving home in the early morning, and by the time they reach the school, it might be 2–3 h after the drug intake time. The drug is still persistently effective in school for 6 h, but as the drug concentration decreases, the likelihood of diversion might also occur, especially during the late afternoon or evening.

Our results demonstrated a much more observable and significant reduction in movement analysis through pixel subtraction. Thus, this new approach can provide better evaluation of the effects of therapy on patients with ADHD based on the fact that the average movement among children with ADHD who had received 1 month of treatment was significantly reduced. The advantage of this method is that it can detect every movement of the subject that has been captured using the video recorder [7]. This suggests that improvement in ADHD symptoms is objectively measurable through the pixel subtraction method, potentially providing an alternative to SNAP-IV.

We further compared the behavioral changes among the children who received 1 month of Ritalin or Concerta treatment through the traditional and movement analysis methods. We observed that a larger proportion of children exhibited improvement in terms of movement reduction of more than 25% through motion analysis. As mentioned earlier, regardless of the treatment prescribed, children with ADHD who had received 6–12 months of treatment reported a symptom reduction of 25% from baseline, as measured using the Vanderbilt ADHD Diagnostic Rating Scale [4]. By contrast, parents and teachers reported that fewer children exhibited improvement in terms of symptom reduction of more than 25%. This may be because the observation period of parents or teachers is too short. However, movement analysis can evaluate therapeutic effects in the early stages of treatment. In addition, our proposed method produces more reliable data as it is measurable and objective.

Another issue is that in movement analysis, only hyperactivity or a combined ADHD subtype (i.e., only patients exhibiting high motor activity) can be analyzed. With its advantage of detecting movement changes, it provided consistent monitoring for hyperactive or combined subtypes of ADHD; this movement analysis can be viewed as being superior to the traditional method. Notably, ADHD-C and ADHD-H subtypes are predominant, with a combined prevalence of 78.0–81.7%, which is much higher than that of ADHD-I (18.3–22.0%) [19,20,21]. Hence, our proposed method can be used for tracking therapeutic effects in most, but not all, patients with ADHD.

In our study, a few issues were noted when applying the movement analysis. First, the option of clothing color was our main concern. We could only choose participants who wore solid plain-colored clothes because clothing texture or color affected the pixel subtraction value. Striped or contrasting-color clothing increased the light field area, thus increasing the movement value. Second, our main focus in movement analysis was the children. It was imperative that the caretaker or parent should not block the view of the video recorder.

Our study has some limitations. Our cohort primarily comprised patients with ADHD-C and with an unbalanced sex distribution; thus, our results may not be representative of all ADHD subtypes and both genders. The small sample size (*n* = 25) is a major limitation in this study. In Table 1, even a reduction of 25.07% (above the cut-off for improvement) is not significant. More ADHD patients should be enrolled in further study to increase the reliability of our results. The movements of patients with ADHD in the consulting room may be affected by non-pharmacological factors, such as food intake on the day of assessment, sleep quality before each visit, and familiarity with the consulting environment. Future studies should include a questionnaire to investigate the relationship between these confounding factors and children’s movements. Although this study showed a 39% reduction of movements as compared to a 16–20% reduction of ADHD-symptoms as rated by parents and teachers, further study should be performed to investigate the relationship between movement values and functional impairment. Furthermore, the camera position and angle are important components in movement analysis. Different camera angles or positions can result in different shooting views and a possibility of failing to detect the overall movement of body parts, leading to inaccuracy or bias in movement analysis. Future studies should use two or more cameras mounted at different locations in the room to overcome this issue.

## 6. Conclusions

In the present study, we demonstrated that pixel subtraction of video images is an objective and quantitative method of evaluating ADHD medication efficacy in the early stage. We suggest that the method could be clinically applicable for managing other behavior/movement disorders of children.

## Figures and Tables

**Figure 1 ijerph-19-03163-f001:**
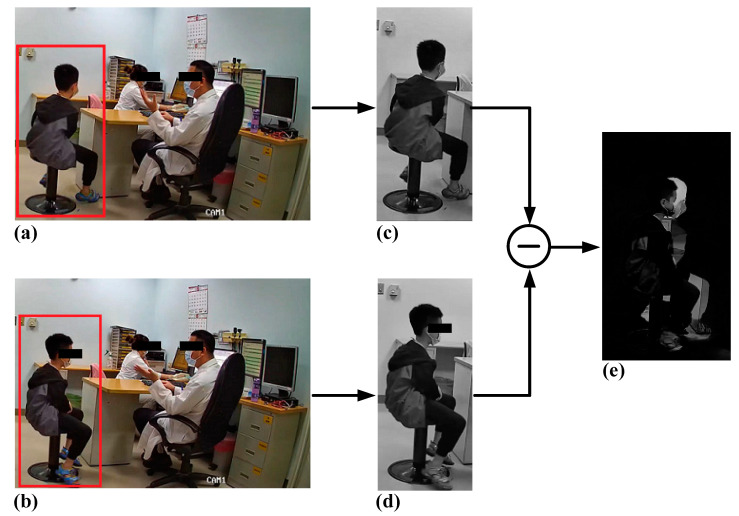
Flow chart of the pixel subtraction method. (**a**) First input color image. (**b**) Second input color image. (**c**) First grayscale image. (**d**) Second grayscale image. (**e**) Subtracted image.

**Figure 2 ijerph-19-03163-f002:**
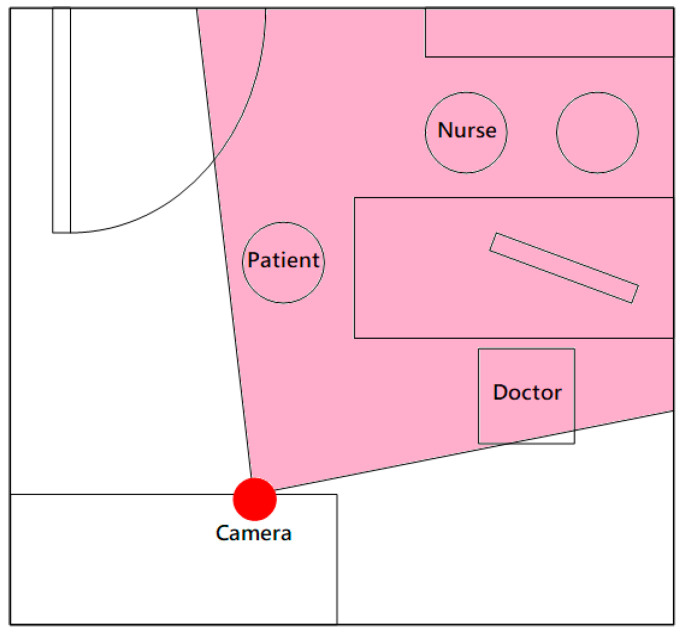
Video recorder’s positioning and view in the consultation room.

**Figure 3 ijerph-19-03163-f003:**
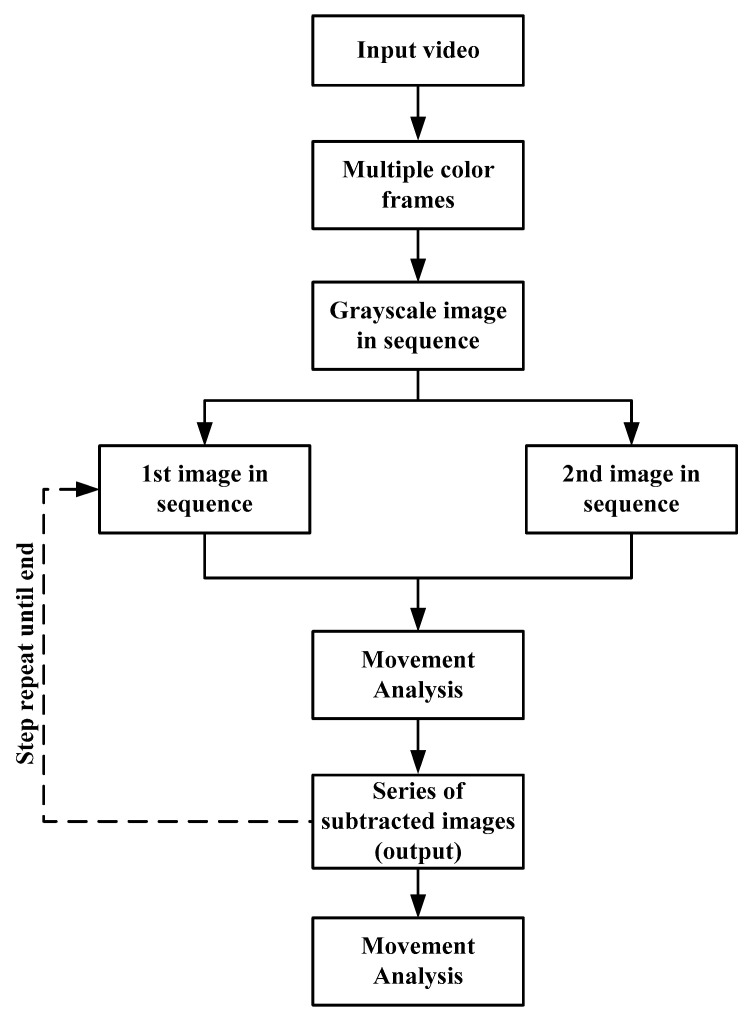
Flowchart of the movement analysis conducted using the pixel subtraction method.

**Table 1 ijerph-19-03163-t001:** Comparison of SNAP subscale scores in 25 patients with ADHD before and after treatment.

Subscales	Before Treatment	After Treatment	Reduction Rate (%)	*p* Value
Inattentiveness (P)	16.44 ± 4.95	13.68 ± 5.12	16.78	0.066
Hyperactivity (P)	15.00 ± 4.67	10.96 ± 5.55	26.93	0.002 *
Oppositional (P)	12.28 ± 5.33	10.44 ± 5.74	14.98	0.189
Inattentiveness (T)	15.40 ± 5.85	14.92 ± 6.35	3.11	0.722
Hyperactivity (T)	13.24 ± 8.69	9.92 ±7.75	25.07	0.057
Oppositional (T)	9.44 ± 6.24	7.00 ± 6.52	25.84	0.048 *

* *p* < 0.05. P: parents. T: teacher.

**Table 2 ijerph-19-03163-t002:** Comparison of the average movement value before and after treatment of 25 patients.

Before Treatment (%)	After Treatment (%)	Reduction Rate	*t*	*p*-Value	95% CI (%)
0.9940 ± 0.7556	0.6036 ± 0.4405	39.27%	3.883	0.001	(0.1829, 0.5979)

**Table 3 ijerph-19-03163-t003:** Treatment outcome comparison in 25 children with ADHD using SNAP scores and movement analysis.

			<25% Reduction (*n*)	>25% Reduction (*n*)
SNAP-IV	Parent	Inattentive	15	10
		Hyperactive	14	11
		Opposition	16	9
	Teacher	Inattentive	18	7
		Hyperactive	11	14
		Opposition	11	14
Movement value			7	18

## Data Availability

Research data are not shared due to privacy restrictions.

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
