# Peer review of "Early and Objective Evaluation of the Therapeutic Effects of ADHD Medication through Movement Analysis Using Video Recording Pixel Subtraction"

_ijerph, 2022, doi:10.3390/ijerph19063163_

Round 1

Reviewer 1 Report

This study recruited 25 patients with ADHD and compared the evaluation of the therapeutic effects of ADHD medication using traditional ADHD assessment tool (SNAP questionnaire) and the average movement value (AMV) assessed using the pixel subtraction technique. The authors concluded that the pixel subtraction method can serve as an objective and reliable evaluation of the therapeutic effects of ADHD medication in the early stage. This study is interesting and the results were well-presented. I have some comments about the manuscript.

1.    How much were the dose of Ritalin or concerta in use? And the proportion of doses and body weight? The dosage in use may prominently affect the behavioral change of children ADHD. Moreover, how researchers monitor the drug compliance of the participants? 
2.    Using the SNAP scores, both parent- and teacher usually scored based on behavior observation within one month medication treatment. However, the pixel subtraction technique evaluated the movement in each video within 3 h of medication administration. The AMV assessed the acute effect of drug administration rather than effects of ADHD medication in the early stage. 

3.    How long was the video recorded in this study? Was the clinical setting or time structure standardized? For example, would the same doctor and nurse appear during the video records? What kind of task (i.e., conversation, reading or just observation) was performed during the video records? In my opinion, AMV only assessed the hyperactivity/impulsivity symptoms of ADHD, and inattention symptoms were unable to be captured by AMV.

4.    The authors reported that the AMV decreased in 18 patients (72%) compared with only 44% and 56% of patients based on parent- and teach-er-reported hyperactivity subscale scores. The readers would like to know the correlation between AMV scores and SNAP scores. And how was the consistency between the treatment response evaluated using AMV and SNAP?

5.    The sample size was small, and sex distribution was unbalanced (male: female = 23:2). In addition. the small sample size reduced the statistical power to analyze whether the accuracy using pixel subtraction technique could be generalized into both genders or different ADHD subtypes. 

Reviewer 2 Report

Authors do not mention the possibility that the described variation in movement rate can derive from factors different from pharmacological effect (e.g. during the second visit children are more familiar with the context). 

Authors give no information about the stability of movements rate in normal conditions (both in typical and in ADHD children) nor of the lenght of the time of video recording, 

Reviewer 3 Report

This article concerns the investigation of using movement analysis based on pixel subtraction as a means to evaluate effects of ADHD medication. The study includes 25 participants (aged 5-16 years) and studies effects of medication after 1 month of treatment. Findings indicate a 39% reduction of movements as compared to a 16-20% reduction of ADHD-symptoms as rated by parents and teachers. The topic of finding objective markers of improvement is timely, the study has its merits and is generally well written. However, I recommend rejection of the manuscript based on the following:

  1. The small sample size (N=25) and the wide age range are major drawbacks of the study. Hyperactivity declines across age, so as a minimum age needs to be taken into consideration in the analyses, and I am not sure that the small sample size allows for that. It would have been better to start with a more homogenous sample. Also , the study lacks power to detect important aspects of symptom change. For instance, in table 2, even a reduction of 25.07% (above the authors cut-off for improvement) is not significant. Did the authors conduct a power analysis, and what effect sizes can be detected?
  2. The authors use 25% as a cut-off for improvement. This cut-off is not explained. It seems arbitrary and nothing in the manuscript justifies using the same cut-off for symptoms and movements as measures by the pixel subtraction. Further, the measure should preferably be validated against functional improvement. As for now, there is no evidence that the reduction in movements has a clinical value for the child or his/her environment. ADHD is after all a disorder connected to functional impairment.

Minor comments:

  1. To aid the reader I believe the method of using pixel subtraction could be described more clearly in the introduction. As a naïve reader, I didn’t understand the processes until I got to the methods section. Perhaps consider move Figure 3 to the introduction?
  2. The information in Table 1 is covered in the text and could therefore be removed.

Round 2

Reviewer 1 Report

The authors have adequately addressed my questions and the manuscript is substantially improved.

Reviewer 3 Report

This is the second review of an article concerning the investigation of using movement analysis based on pixel subtraction as a means to evaluate effects of ADHD medication. The study includes 25 participants (aged 5-16 years) and studies effects of medication after 1 month of treatment. Findings indicate a 39% reduction of movements as compared to a 16-20% reduction of ADHD-symptoms as rated by parents and teachers. The topic of finding objective markers of improvement is timely and the study has its merits and is generally well written. Unfortunately, not all my points have been addressed satisfactory and I can therefore not recommend acceptance of the manuscript:

  1. I addressed the small sample size (N=25) as a major limitation. The authors provide a power analysis proposing that they indeed had a sufficient sample size. I find this contradictory, as their set cut off (25%) for clinical improvement was not statistically significant. To me this is a clear indication of an underpowered study. The authors could use Bayesian statistics to investigate the matter further (that is, if they wish to say that teacher-rated hyperactivity did not decline statistically rather than them not having the power to detect this reduction).
  2. My second point concerned age. I would recommend the authors to provide mean and standard deviation for age and provide correlations between age and improvement for descriptive purposes, if not wanting to include age as a covariate in the analyses.
  1. The authors use 25% as a cut-off for improvement and have now provided a rationale for this, based on guidelines from Wolraich. Even though this might be a functional way of addressing symptom reduction based on symptoms, there is no evidence that the same cut off can be applied to movement reduction. This should at the least be discussed as a limitation and future studies should include relations between movement reduction and functional impairment.
